SFTA-Net: a self-supervised approach to detect copy-move and splicing forgery to leverage triplet loss, auxiliary loss, and spatial attention

Alabrah Amerah aalobrah@ksu.edu.sa
Department of Information Systems, College of Computer and Information Sciences, King Saud University , Riyadh , Saudi Arabia
Coelho Paulo Jorge
Electronic publication date: 2025 Apr 16
Publication date: 2025
Volume: 11
Electronic Location ID: e2803
Received 2024 Aug 16; Accepted 2025 Mar 13
Copyright: © 2025 Alabrah
Copyright year: 2025
Copyright holder: Alabrah
License: This is an open access article distributed under the terms of the Creative Commons Attribution License, which permits unrestricted use, distribution, reproduction and adaptation in any medium and for any purpose provided that it is properly attributed. For attribution, the original author(s), title, publication source (PeerJ Computer Science) and either DOI or URL of the article must be cited.
License URL: https://creativecommons.org/licenses/by/4.0/

Keywords: Auxiliary loss, Copy-move, Digital imagery, Forgery detection, Splicing, Self-supervised, Spatial attention

Funding: Researchers Supporting Project RSP2025R476 King Saud University, Riyadh, Saudi Arabia This research was supported by the Researchers Supporting Project (RSP2025R476), King Saud University, Riyadh, Saudi Arabia. The funders had no role in study design, data collection and analysis, decision to publish, or preparation of the manuscript.

==============================
Image forgery is an increasing threat, fueling misinformation and potentially impacting legal decisions and everyday life. Detecting forged media, including images and videos, is crucial for preserving trust and integrity across various platforms. Common forgery techniques like copy-move and splicing require robust detection methods to identify tampered areas without explicit guidance. The previously proposed studies focused on a single type of forgery detection utilizing block-based and key-point feature selection-based classical machine learning (ML) approaches. Furthermore, applied deep learning (DL) methods only focus on deep feature extraction without considering the focus on tampered regions detection or any domain-specific loss. Therefore, this study addresses the aforementioned challenges by proposing a lightweight DL approach, a self-supervised, triplet and auxiliary losses-based forgery detection network (SFTA-Net), featuring a self-guidance mechanism for detecting tampered regions with a commutative loss within images. The SFTA-Net method is proposed to classify forged and original photos belonging to copy-move and splicing forgeries. To effectively analyze the added components in the proposed model, three experiments were conducted, one with a self-guided (SG) head-based convolutional neural network (CNN), a second with SG-head and auxiliary loss, and a third one with SG-head auxiliary loss and triplet losses-based CNN. For experimentation, CASIA 1.0 and CASIA 2.0 datasets were used with 80-10-10% train-validation and test ratios. The testing results achieved on CASIA 1.0 were 95% accuracy and 97% accuracy on the CASIA 2.0 dataset. To prove the approach’s robustness and generalization, the CASIA 2.0-trained weights were used to test on the MICC-FC2000 dataset and yielded limited results. To improve the results, fine-tuning was performed on CASIA 2.0 weights utilizing the MICC-FC2000 dataset which achieved 98% accurate results. Our findings demonstrate that the SFTA-Net surpasses the baseline ResNet18 model and previous state-of-the-art (SOTA) methods. Overall, our SG approach offers a promising solution for detecting forged images across diverse real-world scenarios, contributing to the mitigation of image forgery and preservation of trust in digital content.

Introduction

Digital images are the most prevalent form of data communication on social media platforms. The use of mobile phones and image-capturing tools has become effortless in today’s world. Likewise, image editing tools are widely available and accessible. These image editing tools have increased the amount of fake or forged data spread across digital communication mediums. Therefore, using forged images, videos, and audio to form an opinion has become risky (Bhowal, Neogy & Naskar, 2024). These forged images are highly susceptible to deceitful uses, including manipulation which can impact individuals and communities not only financially but morally (Walia et al., 2021).

Image forgery denotes the action of manipulating a few of the meaningful or useful traits of a digital image to showcase false information (Muniappan et al., 2023). The ultimate goal behind forged digital imagery is to create serious consequences, particularly those that affect journalism, judicial, and legal proceedings in courtrooms (Roy, Mohiuddin & Sarkar, 2024). Image forgery can lead to misjudgment in cases, as well as in literature works, science, medicine, and the military. For humans, detecting a forged image with the naked eye is impossible.

Hence, a variety of methods and instruments are available that can help in image forgery detection. These methods include solutions based on deep learning (DL) algorithms (Elaskily et al., 2023). However, to improve these methods, the detection of forged digital images has become a significant research study domain. In the literature on image forgery detection, many state-of-the-art (SOTA) methods have been proposed (Jing & Tian, 2020; Meena & Tyagi, 2019). These methods include two main types of image forgery, copy-move, and splicing. In the copy-move forgery method, the regions of the images are copied to the same given slice, which increases new content in the original sample (Koul et al., 2022). In contrast, the splicing technique merges the region of one image into another, resulting in a fabricated image (Khudhair et al., 2023). Copy-move forgery is not applied as extensively as the splicing forgery technique. In copy-move forgery, the object is hidden or duplicated in the given sample (Verma & Singh, 2024) while, in the splicing technique, the region is copied from one sample and pasted in another to create a new sample. In short, in both techniques, there is no overall sign that the image has been altered; however, statistics may have changed (Al-Shamasneh & Ibrahim, 2024). In previously proposed methods, two common techniques for image forgery detection have been proposed: the active and passive approach (Jaiprakash et al., 2020). In active/intrusive detection methods, an image with watermarks is inserted and the difference is taken out to manipulate this approach. In contrast, passive methods compute, the statistical features are computed between the testing sample and pre-fed information of the targeting sample (Baumy et al., 2022).

A recent review by Verma & Singh (2024) on forgery detection claimed that many intelligent methods have been proposed for digital image forgery detection including fragile and semi-fragile image compression, image resampling, and a few other techniques. However, even with all these methods, a research gap still exists, namely because these methods belong to a single type of forgery detection, and only a few have been proposed as solutions for both types. Another research problem is that the applied approaches give block-based forgery detection solutions that are not robust enough to be effective on geometric transformations. Likewise, key-point features-based techniques are not robust. If block-based and key point feature-based techniques are composed then the approach becomes complex in computation. Therefore, a robust and generalized solution is necessary for copy-move and splicing forgery detection, and additionally, a lightweight solution is needed. To employ these forgery detection methods, DL solutions have been proposed that need abundant data for training, which is also another problem (Baumy et al., 2022). Therefore, to decrease the spread of fake information and detect image forgery, an intelligent, robust, and lightweight method is needed. The core focus of this study is to enhance the passive forgery detection (Copy-Move & Splicing) technique by providing a robust DL solution. The derived contributions of this study are listed below; A generalized SG convolutional neural network (CNN) model is proposed for copy move and splicing image forgeries without using labeled masks to detect the tampered and original areas

In the proposed network, the auxiliary head is added to incorporate any missing information loss during CNN layers

A commutative loss is proposed in a combination of classification loss, auxiliary loss, and triplet loss to increase the distance between tampered and original image features

Based on essential components, a lightweight CNN model, namely the self-guided triplet and auxiliary losses-based forgery detection network (SFTA-Net) is proposed which identifies the tampered regions with a self-supervision-utilizing attention head, with auxiliary information and the triplet loss involvement.

This article consists of the following sections: “Related Work” delivers a comprehensive review of the related literature, covering all terms and definitions of relevant forgery detection approaches, DL, and various contributions to research. “Methodology” details the methodology adopted for dataset preparation, model training, and evaluation. “Pseudo Code for Network Forward Pass and Loss Computation” and “Results and Discussion” reviews the experimental results and their analysis, followed by a discussion. “Conclusion” concludes the article with key findings, limitations, and future research directions.

Related work

In this section, a brief literature review on digital image forgery detection is described. Both types of image forgery, SOTA deep learning, and other approaches have been described in the literature. Alencar et al. (2024) investigated the multi-stream novel CNN method in which two heads extracted certain subsets belonging to the proposed dataset. Furthermore, employing a third head was proposed in the same CNN on the unchanged sample. It utilized four datasets to make a new dataset covering passive forgery types. The reported results were 89.50% accurate on validation data. However, it did not provide any implications related to the tampered region in the image, which limited its practical implications.

Another study by Chakraborty, Chatterjee & Dey (2024) proposed a hybrid approach that combined traditional handcrafted features and a dual-faceted approach of CNN to detect tampered images. It further utilized an error level analysis (ELA), and noise residuals from the spatial-rich model. The CASIA 2.0 dataset was used to train and test on a hybrid CNN model and achieved 98.55% accuracy. Al-Shamasneh & Ibrahim (2024) used a two-stage method to detect forged images. This method leveraged mathematical functions, namely the sonine function, based on convex features that are extracted to identify the texture and structural information of an image. Second, it added deep features and applied a support vector machine (SVM) as a classifier, resulting in 98.93% accuracy on the CASIA 2.0 dataset. However, it includes structural and deep features but lacks a DL-based, robust, end-to-end approach. This study employed DL techniques, including ELA and CNNs, to detect counterfeit photographs, and achieved high accuracy using an ELA-CNN model (99.87%).

A two-stage method of forgery detection was proposed (Ye et al., 2022). In this method, forged areas are detected using a single feature-based method. The method fuses the scale-invariant feature transform (SIFT) and Hu features to detect the forged regions. Subsequently, threshold-based histogram of oriented gradient (HOG) features were extracted for forged area identification. Results showed superior performance, with 99.01% and 98.5% accuracy achieved on the MICC-F220 and MICC-F2000 datasets, respectively, with enhanced robustness on the COMOFOD dataset.

Ravikumar et al. (2024) gave a comparative analysis-based forgery detection solution. It included the ELA and CNN models with the comparative performance of the VGG-16 model. It also utilized the CASIA 1.0 dataset which contains only one type of forged image. The results indicate that the VGG-16 model achieved higher performance with 85% precision, recall, and F1-score. This study was limited to only one type of forgery detection and also did not provide any efficiency in identifying tampered patterns present in the images. Another comparative study by Mallick et al. (2022) utilized the CASIA 2.0 and NC2016 datasets and classified the data using ELA, VGG-16, and VGG-19 CNN models. The achieved testing accuracy accuracies were 70.6%, 71.6%, and 72.9% respectively. There was no forged region-specific or spatially-guided module added to improve the performance of the forgery detection approaches. Simply, the fine-tuning of classification models was reported.

Another study stated that, with the increase of image editing tools, identifying the tampered images becomes challenging (Alahmadi et al., 2017). Therefore, a novel passive forgery detection method was proposed that combines local binary patterns (LBP) features and discrete cosine transform (DCT) to identify the copy-move and splicing categories of forgery. Discriminative localized features are extracted using a chrominance component from the 2D DCT in the LBP space, followed by detection utilizing the SVM method. The experimental results proved the superiority of the proposed method.

Similarly, Khalil et al. (2023) employed DL approaches by utilizing transfer learning methods to detect both types of image forgeries. It involved comparing the original image with its compressed version and the difference between both types of images. It utilized eight pre-trained weights while the foremost performance-gaining method was MobileNetV2 with 94.69% accuracy, 0.94 F1, 94.21% precision, 94.74% recall, and 0.95 AUC on the CASIA 2.0 dataset. Likewise, another approach (Ali et al., 2022) was applied that focuses on the double image compression technique. It trained a CNN-based method using the original and compressed images. The lightweight model achieved an overall validation accuracy of 92.23% using the CASIA 2.0 dataset with an 80-20 (training-testing) split. This method claims to mitigate the limitation of CNN-based existing techniques for both types of forgeries. Pre-trained model feature utilization is common in computer vision. Likewise, this study (Qazi, Zia & Almorjan, 2022) proposed a ResNet50v2-based approach for detecting image splicing by leveraging YOLO weights for enhanced performance. Two datasets of CASIA were used in experiments, and CASIA 1.0 and CASIA 2.0 datasets achieved 99.3% accuracy with transfer learning and 81% without it. Results highlighted the system’s robustness and superiority over existing methods. A custom network called MiniNet proposed by Tyagi & Yadav (2023), is lightweight and fully convolutional. It was trained on 140 k images belonging to real and fake faces and it utilized CASIA datasets. Ablation studies on SOTA CNNs explored the impact of self-attention, positional encoding, and model depth, showcasing MiniNet’s minimal architecture and robust performance across various datasets. Another DL model-based approach was applied utilizing multi-scale residual connections. It was designed to detect multiple types of image manipulation operations and includes three stages preprocessing, hierarchical-level feature extraction, and classifying modules. These collectively showed promising classification results with 97.07% on the BOSSBase dataset and 97.48% accuracy utilizing the Dresden dataset. Zhao & Tian (2022) used deep learning-based algorithms when proposing a model for splicing forgery detection namely a multiscale fusion model. By modifying the structure of the network, features like dilated convolution, maximal pooling, and pyramid pooling module, the forgery detection performance of MobileNetv2 architecture was enhanced. It achieves high precision and F-score on the CASIA and COLUMB datasets with minimal computational requirements. The classification results of CASIA datasets were 0.91 PREC, 0.80 REC, and 0.832 F1. Likewise, on the COLUMB dataset, the testing PREC 0.964, REC 0.852, and F1 0.881 scores were reported.

Another study by Cozzolino & Verdoliva (2019) used the “Noiseprint” method to capture fingerprint-like structures as well as a Siamese network for forensic analysis. This focused on enhancing the artifacts while suppressing high-level representations to identify the forged areas using pairs of patches of images. It elaborated on the robust alternates of detecting and localizing image forgeries by overcoming major challenges of noise interference and dataset constraints in PRNU-based methods. Transformer-based approaches were also proposed previously, in which an attention mechanism is used for forensic analysis. Such as “TruFor”, a fusion approach proposed by Guillaro et al. (2023), which incorporates high and low-level traces by declaring the pixel-level localization and a reliability map. It provides a global score to provide confidence against false alarms. The multi-scale feature analysis module provides reliable decision-making under complex conditions. The results indicate that the proposed method outperforms the SOTA methods in detecting manipulated images and identifying forged areas, making it a more practical approach. The scale, size, and orientation invariance is an open challenge in the computer vision domain, particularly in the rotation invariant approach in copy-move forgery, which uses the fast algorithm (Cozzolino, Poggi & Verdoliva, 2015a). It provides localization using field techniques and rotation-invariant features. Experimental evaluations demonstrate significant advancements in localization accuracy and speed compared to existing methods. Another work (Cozzolino, Poggi & Verdoliva, 2015b) utilized the co-occurrence of image residuals for splicing detection without prior knowledge, leveraging local feature extraction to identify spliced areas. The proposed approach achieves reliable splicing localization, even with limited training data. The results have proven robustness in both supervised and unsupervised settings.

Agarwal, Walia & Jung (2024) used texture features namely LBP features, which were orientation invariant, and such as discrete cosine transform were extracted to detect copy-paste and splicing concurrently. This study employs an SVM classifier utilizing Accelerated-KAZE (AKAZE) features that helped in the identification of replicated regions, and, ultimately, were used for copy-paste detection. Results demonstrate highly improved performance of the approach as compared to the previously used methods on Extended IMD2020, CASIAv1.0, and CASIAv2.0. Furthermore, the nature of the approach permits concurrent detection of both forgeries of any type, ultimately enhancing its utility in the domain of digital image forensics.

In related studies about forgery detection Table 1, different kinds of handcrafted features, transfer learning, dual branch, and fusion-based DL models have been proposed and achieved promising results as well. However, these studies did not focus on tampered areas-based forgery detection in an SG manner. The self-supervision makes the applied approach a more robust and generalized way to identify the tampered and original images in any kind of forged imagery. Similarly, the forged and original images need to be separate which can be done using pairwise distancing on forged and original images. It can more efficiently cluster tampered and original images.

Table 1 Employed methods, datasets, and their achieved results to develop forgery detection systems.

Reference	Year	Method	Dataset	Results	
Alencar et al. (2024)	2024	Multi-head CNN model	Four datasets based on, a new dataset	Accuracy = 89.50%	
Chakraborty, Chatterjee & Dey (2024)	2024	Hybrid CNN model includes hand-crafted features and CNN	CASIA 2.0	Accuracy = 98.55%	
Al-Shamasneh & Ibrahim (2024)	2024	Texture and structural features fused with deep Features and SVM classifier	CASIA 2.0	Accuracy = 98.93%	
Ravikumar et al. (2024)	2024	ELA-CNN and VGG-16 CNN	CASIA 1.0	Precision. Recall and F1 = 85.0%	
Mallick et al. (2022)	2023	ELA, VGG-16 and VGG-19	CASIA 2.0 and NC2016	Accuracy = 70.6%, 71.6%, and 72.9%	
Khalil et al. (2023)	2023	Transfer Learning on eight pre-trained weights with original and compressed images differences	CASIA 2.0	ACC = 94.69%, F1 = 0.94, PREC = 94.21%, REC = 94.74%, and AUC = 0.95	
Ali et al. (2022)	2022	Double compressed images and difference based CNN model	CASIA 2.0	ACC = 92.23%, PREC = 0.85, REC = 0.97, and F1 = 91.08%	
Zhao & Tian (2022)	2022	Multi-scale feature fusion CNN model	CASIA and COLUMB	CASIA PREC = 0.91, REC = 0.80, F1 = 0.832 COLUMB PREC = 0.96, REC = 0.85, F1 = 0.881 scores	

The use of triplet loss is incorporated where contrastive inclusion is needed in guided or self-supervision tasks. A different study used “dummy-triplet loss,” which simplifies triplet loss by optimizing the fixed positive and negative vectors instead of semi-hard triplets (Beuve, Hamidouche & Deforges, 2021). Likewise, a dual triplet loss combination is used to identify between random and mature forgeries (Wan & Zou, 2021). This method relies on triplet constructions for each forgery type found making it a less generalized approach. The self-supervision has also been proposed in previous studies such as in the self-information attention method (Sun et al., 2022). It added a self-attention module to enhance the features to identify the deepfakes. The high frequency-based noise features are extracted as unique patterns for face forgery detection (Luo et al., 2021). It integrates triplet loss with the incorporation of multi-scaled features and attention mechanisms to improve detection and generalization. In contrast to these studies, the proposed method addresses key limitations by providing a self-supervised learning framework that does not rely on ground-truth tampered image masks, which increases flexibility and applicability to a wide range of forensic tasks. Furthermore, using attention mechanisms that, in combination with triplet loss, focus on critical regions of the image and improve forgery localization, goes beyond the attention modules. The generalizing across different forgeries and datasets is provided by a more robust solution for cross-dataset generalization by addressing the specific dataset overfitting issues present in previous studies. In the proposed method, a commutative loss is incorporated utilizing a multi-loss function approach (cross-entropy, auxiliary, and triplet loss), which is more comprehensive and effective for detecting and localizing forgeries.

Methodology

In this section, the dataset details are discussed and the SG-forgery detection network is explained in detail. The SFTA-Net has been specifically designed to focus on tampered areas via SG-attention loss with auxiliary loss. Furthermore, a triplet loss is added to create more distance between tampered and original images. In this study, the dataset was split up into train, validation, and test as 80-10-10 split ratios. The experiment was performed on core i7, a 10th generation machine with 16GB RAM and 6GB NVIDIA RTX GPU using a Python framework. The experimental setup with parameters is shown in Table 2. The primary steps of the applied method in this framework are shown in Fig. 1.

Table 2 Training settings and parameters.

Parameter	Value	
Epochs	100	
Batch size	32	
Optimizer	Adam	
Learning rate	0.001	
Loss functions	Cross-entropy loss, triplet loss	
Triplet loss margin	1.0	
Input image size	(256, 256)	
Data augmentation	Resize, normalize	
Model architecture	CNN with attention	
Classifier output size	2	
Auxiliary classifier	Yes	

Figure 1 Proposed self-guided network for forgery detection with auxiliary loss and triplet losses.

It illustrates that both types of images are given class labels to the network, and spatial attention is added to them by extracting features from multiple CNN layers. These features are passed through two classification heads, and a triplet loss is also added to reduce the distance between positive and negative samples. Both head losses and triplet losses are collectively back-propagated to update the network. The input images are fed to the network without using their tampered area masks, and self-supervision is applied via SG-block.

The SG-head uses the deep features and calculates a guided vector added with features while backpropagating the network. This way, the SG-block detects potentially tampered and normal regions in the input images. The detailed representation of the applied network is explained in the next section.

Dataset description

In this framework, we utilized the CASIA 1.0, CASIA 2.0 (Dong et al., 2013), and MICC-FC2000 (Amerini et al., 2011) datasets available on the Kaggle website. In these datasets, two directories were mainly available with tampered and original images. In contrast, the masks of tampered regions were also available but were not utilized when proposing a self-supervised method. However, the main task was to identify the tampered and original images using segmentation or classification. The dataset details are given in Table 3.

Table 3 Datasets details used in this study.

Dataset	Class	Train	Test	Validation	Total	
CASIA 1.0	Tampered	735	95	91	921	
Original	641	78	81	800	
Total		1,376	173	172	1,721	
CASIA 2.0	Tampered	4,088	505	530	5,123	
Original	6,003	757	731	7,491	
Total		10,091	1,262	1,261	12,614	
MICC-FC2000	Tampered	580	62	58	700	
Original	1,020	138	142	1,300	
Total		1,600	200	200	2,000	

In CASIA 1.0 model training, 80% of data was used with 1,376 images of both tampered and original classes, whereas during training, the model performance was estimated using validation data. After training, the model, testing was performed on testing data using 172 images of both classes. Similarly, for the CASIA 2.0 dataset, 10,091 images were used that belonged to both types of classes. Testing and validation were performed on almost the same amount of data as on 1,262 images. Each experiment was performed for both datasets including baseline methods and the proposed SFTA-Net method. The CASIA 2.0 dataset-based trained model was tested on the MICC-FC2000, dataset whereas fine-tuning performed on CASIA 2.0 weights utilized the training data of the MICC-FC2000 dataset. Later on, testing was performed on the newly fine-tuned model using the MICC-FC2000 dataset.

SFTA-Net

The SFTA-Net was developed to better consider the tampered region of images and to add some kind of guesses and indications to the model for a better understanding of tampered and original images. For a fair comparison, an ablation study was added without the use of an auxiliary head, and a triplet loss was added to evaluate the performance of SFTA-Net. The attention head was added to the lightweight CNN network, for the feature extraction. The pre-trained weights were trained on big data and have spatially enriched information in them, but were complex in weight and parameters. The ResNet18 weights were used in the experiment showed lower results as compared to the proposed SFTA-Net method. While experimenting, it was considered that some information was lacking due to lack of performance, therefore an auxiliary head was added with a classification head, and attention was added to CNN, namely SFTA-Net. It is lightweight as compared to the ResNet18 backbone and attention-added ResNet18 model.

In SFTA-Net, three convolutional and three ReLU layers are used to extract features. Subsequently, these features have been used in SG head to calculate guesses and indications of tampered and original regions in the images. The SG head uses three feature vectors from the backbone query, key, and value vectors. These query (Q), key (K), and value (V) vectors and features are shown in Eqs. (1)–(3):

(1) Q=Wq⋅X

(2) K=Wk⋅X

(3) V=Wv⋅X.

These weighted vectors are used to add attention score or spatial significance in feature maps. The mathematical formulation is shown in Eq. (4) is formed by applying SoftMax on the dot product of Eqs. (1) and (2).

(4) α=softmax(Q⋅KT).

The normalized vector α contains the probabilities which are calculated using the SoftMax function on the dot product of the transformed key vector and query vector. The dot product is then calculated in between the normalized vector α and value vector and the convolutional features are then added as shown in Eq. (5).

(5) output=X+(V⋅α).

The value vector is copied from convolutional layers and important or valuable indications are added with the help of the calculated normalized vector. However, when we added a value vector-based product, the indications were added to respective places or regions. Finally, the convolutional features and guided vector were summed as input features to the SG head of the applied network. The network layers or architecture are explained in Table 4.

Table 4 Layer details of the SFTA-Net model.

Layer name	Layer type	Input size	Output size	Parameters	
Conv1	Convolution	3 × 256 × 256	32 × 256 × 256	896	
Pool1	MaxPool	32 × 256 × 256	32 × 128 × 128	0	
Conv2	Convolution	32 × 128 × 128	64 × 128 × 128	18,496	
Pool2	MaxPool	64 × 128 × 128	64 × 64 × 64	0	
Conv3	Convolution	64 × 64 × 64	128 × 64 × 64	73,856	
Pool3	MaxPool	128 × 64 × 64	128 × 32 × 32	0	
Spatial_attention	SpatialAttention	128 × 32 × 32	128 × 32 × 32	20,992	
FC1	Linear	131,072	256	33,554,688	
FC2	Linear	256	2	514	
aux_FC1	Linear	131,072	256	33,554,688	
aux_FC2	Linear	256	2	514	

The network architecture is described in Table 4, in which the convolutional and ReLU layers are used to extract features that are given to a spatially guided head which is further applied to multiple operations and outputs the guided vector. The guided vector and convolutional features are then fed to the two classification heads which use the FC layers given the output of original and tampered regions. Given that two classification heads are backpropagated at once after considering the important gradient signals which include both main and auxiliary tasks, this results in a more informative update to the network. Furthermore, triplet loss is added which computes the pairwise distance between positive and negative pairs of anchor. Both losses are used to compute a total loss.

Auxiliary loss

The SFTA-Net proposed includes an SG-attention mechanism used to focus on tampered regions. However, to utilize the auxiliary information that may be lost during classification head backpropagation, two classification heads were added and cross-entropy loss on both heads was computed while the sum of both losses was added to compute the commutative loss, specifically auxiliary loss. The total loss Ltotal as the sum of two cross-entropy losses L1 and L2 can be written as Eq. (6):

(6) Ltotal=L1+L2

where each cross-entropy loss Li (for i=1,2) is defined as in Eq. (7):

(7) Li=−∑c=1Cyclog⁡(y^c).

Here, yc is the true label for class c and y^c is the predicted probability for class c.

Triplet loss

In this loss, an instance is taken from one of the classes as 0 and 1. Then three images are sampled from the given dataset as the anchor (given image), positive (from the same class as the given image), and negative sample from another class as of the given image. The pairwise distance is calculated between positive and negative samples to compute the triplet loss. The mathematical formulation is shown in Eq. (8). Here a denotes the anchor image, p represents the positive image, and n signifies the negative image, while d indicates a distance function (e.g., Euclidean distance), and m is a margin that enforces a minimum separation between positive and negative pairs relative to the anchor. The goal of the triplet loss is to minimize the distance between the anchor and the positive instance while maximizing the distance between the anchor and the negative instance.

(8) L=max(d(a,p)−d(a,n)+m,0).

Combining these, we can express the total loss as Eq. (9).

(9) Ltotal=−∑c=1Cy1clog⁡(y^1c)−∑c=1Cy2clog⁡(y^2c)+max(d(a,p)−d(a,n)+m,0).

Based on this combined loss, the feature information efficiently enhanced the network and improved classification results. However, both of the heads could be used to get the prediction of whether the network was tampered with or was original. In the cumulative loss, three losses were added to effectively improve the model performance on forgery detection. The pseudo-code of the commutative loss and forward-passes-based network is shown in Algorithm 1.

Algorithm 1 Forward pass and loss computation.

Input: Image (x)	
conv1_out←Conv1(x)	
pool1_out←MaxPool(conv1_out)	
conv2_out←Conv2(pool1_out)	
pool2_out←MaxPool(conv2_out)	
conv3_out←Conv3(pool2_out)	
pool3_out←MaxPool(conv3_out)	
spatial_attention_out←SpatialAttention(pool3_out)	
flattened←Flatten(spatial_attention_out)	
fc1_out←FC1(flattened)	
fc2_out←FC2(fc1_out)	
aux_fc1_out←auxFC1(flattened)	
aux_fc2_out←auxFC2(aux_fc1_out)	
Lmain←−∑c=1Cymain,clog⁡(y^main,c)	
Laux←−∑c=1Cyaux,clog⁡(y^aux,c)	
Ltriplet←TripletLoss(anchor,positive,negative)	
Ltotal←Lmain+Laux+Ltriplet	
Return Ltotal	

Pseudo code for network forward pass and loss computation

The algorithm is completed in 17 network training steps. This includes the input images as x being passed through convolved layers and a pooling layer, and then these features or high-level image representations are passed to the module (SG) to obtain focusing features that indicate the tampered regions present in the image. Based on the output of the SG module, the two classification heads are utilized with fully connected layers. Later on, cross-entropy losses on both fully connected layers are added in steps 12 and 13 which are the outputs of FCs, and given to cross-entropy losses in steps 14 and 15. Subsequently, the input image-based positive and negative samples are added and triplet loss is computed at step 16. Lastly, the losses are summed up to utilize a commutative loss.

Results and discussion

For experimental purposes, the simple ResNet18 network is first applied to determine the need for a spatially guided head to identify the tampered and original regions. The results of ResNet18 were not so promising, and the ResNet18 weights were big compared to the proposed SFTA-Net model. For experimentation, the datasets were split up into train, test, and validation as 80, 10, and 10 ratios, respectively to get the training and performance measures of applied forgery detection models.

In Table 5, the results of all three added components in SFTA-Net development are shown individually, and in the combined form on all three datasets. When we analyzed the self-attention mechanism and classification loss-based SG-Net model, it gave only 63% accuracy while F1 remained very low, resulting in higher misclassification among both classes. The second component, with auxiliary loss, was added to include information that may have been lost while in pooling layers. Although the results did not improve in the case of CASIA 1.0, they improved in the case of the CASIA 2.0 dataset. However, to further improve the results, the triplet loss was added which utilized the features of positive and negative anchors and added differences in weights to get closer to similar class instances and further from the opposite class instances. Therefore, a combined combination improved the overall results and reached up to 95% accuracy, precision, recall, and F1 scores. Likewise, in the case of CASIA 2.0 testing data, the SFTA-Net model gradually increased results with the inclusion of auxiliary loss and triplet losses. This resulted in 97% precision, recall, f1, and accuracy scores. To check generalization, the final weights of CASIA 2.0 datasets were used to further fine-tune only 10 epochs instead of training from scratch. The final results of MICC-FC2000 resulted in higher results as 95% precision, 100% recall, 97% precision, and 98% overall accuracy. However, the results without fine-tunning on MICC-FC2000 are also shown in Table 5, which showed poor results, whereas with slight fine-tunning only 10 epochs in CASIA-2.0 trained weights improved the model generalization and showed final results in MICC-FC2000 as 95% precision, 100% recall, 97% precision, and 98% overall accuracy. The achieved results of all datasets are compared with SOTA methods, and proven as improved results. The detailed tabular comparison is given in Table 6.

Table 5 Ablation study: performance of SG-Net, SG-Aux-Net, and SFTA-Net models on CASIA 1.0, 2.0 and MICC-FC2000 datasets.

Dataset	Model	PRE (%)	REC (%)	F1 (%)	ACC (%)	
CASIA 1.0	SG-Net	65.0	56.0	51.0	63.0	
SG-Aux-Net	60.0	55.0	49.0	57.0	
SFTA-Net	95.0	95.0	95.0	95.0	
CASIA 2.0	SG-Net	75.0	72.0	68.0	68.0	
SG-Aux-Net	75.0	74.0	69.0	69.0	
SFTA-Net	97.0	97.0	97.0	97.0	
MICC-FC2000	SFTA-Net (w/o Fine-Tunning)	60.0	59.0	47.0	47.0	
MICC-FC2000	SFTA-Net	95.0	100	97.0	98.0	

Table 6 Results for baseline and SOTA methods are reported as per their respective original articles on CASIA 1.0, 2.0, and MICC-FC2000 datasets.

Reference	Dataset	Methodology	PRE (%)	REC (%)	F1 (%)	ACC (%)	
Ali et al. (2022)	CASIA 2.0	Double compression, CNN	85	97	91.08	92.23	
Khalil et al. (2023)	CASIA 2.0	Transfer learning, compression	94.21	94.74	94.00	94.69	
Muniappan et al. (2023)	CASIA 1.0	CNN	–	–	–	79.0	
CASIA 2.0	CNN	–	–	–	89.0	
Proposed	CASIA 1.0	SFTA-Net	95.0	95.0	95.0	95.0	
CASIA 2.0	SFTA-Net	97.0	97.0	97.0	97.0	
MICC-FC2000	SFTA-Net	95.0	100.0	97.0	98.0	

In Table 6, the first three compared studies were explained above in section 2 whereas they were included in the table due to being the same dataset. Due to computational resource constraints, the reported results for other methods are stated in their original articles. It allowed us to evaluate methods already benchmarked on chosen datasets but limited the ability to include methods that have not reported results for these datasets. Although this limitation affects the comprehensiveness of the comparative analysis, we believe that our findings remain valuable for highlighting the performance of the proposed method within the scope of available benchmarks.

In the first compared study, the double compression technique was used whereas the CNN model was used to test on the CASIA 2.0 dataset. Likewise, the second comparative study also used compression and transfer learning on the CASIA 2.0 dataset, giving 94.69% accuracy on both classes. The third comparative study utilized the CNN model and applied it on both CASIA 1.0 and CASIA 2.0 datasets, reaching up to 79.0% and 89.0% testing accuracy, respectively. In comparison to these studies, the SFTA-Net model contains spatial attention, an added to the auxiliary head, and a triplet loss, which leads to improved results. The CASIA 1.0 and 2.0 showed improvement when compared to SOTA methods. However, to show the robustness and generalization of the proposed method, the CASIA 2.0 dataset-based trained weights were used to test on a different dataset, the MICC-FC2000. The reliance on fine-tuning highlights the necessity of addressing cross-dataset generalization challenges in future work. While the proposed method showed SOTA performance on CASIA datasets (1.0, 2.0) and improved performance on MICC-FC2000 with fine-tuning, direct application without adaptation remains a limitation. Leveraging the CASIA 2.0 datasets-based trained model learning, lightweight 10 epochs-based fine tunning was applied on the MICC-FC2000 training set and resulted in 98% accuracy and the same precision, recall, and F1 scores. The ROC curves were also played an important role in estimating the model performance of positive and negative classes, particularly when datasets are imbalanced. Therefore, to tackle and visualize the model performances in all three experiments, the testing ROC curves are calculated and shown in Fig. 2.

Figure 2 Testing ROC curves (A) CASIA 1.0 SG-Net, (B) CASIA 2.0 SG-Net, (C) CASIA 1.0 SG-Aux-Net, (D) CASIA 2.0 SG-Aux-Net (E) CASIA 1.0 SFTA-Net, (F) CASIA 2.0 SFTA-Net, (G) pre-trained CASIA 2.0 model testing on MICC-FC2000 dataset (H) fine-tuned CASIA 2.0 model on MICC-F2000 dataset.

In Fig. 2, the right column shows all three experiments’ ROC curves for CASIA 1.0 datasets and the left column shows all three experiments’ testing ROC curves of CASIA 2.0 dataset. From analyzing the curves, we found that the SG-Net experiment showed similar behavior with a precision of a 0.67 score and 0.83 on CASIA 1.0 and CASIA 2.0 respectively. Similarly, the performance of 2nd experiment remains similar with a slight decrease instead of 0.65 and 0.82 scores on CASIA 1.0 and CASIA 2.0 datasets, respectively. However, in the last experiment based on the proposed model, the 0.99 ROC score was achieved on both datasets. The SFTA-Net is not only theoretically or implementation levels, but was also able to pick up on tampered areas and visualize the SFTA-Net-based detection, the feature maps are shown in Fig. 3. Figures 2G and 2H clearly illustrate the existence of robust features when using cross datasets for forgery detection with a higher true positive rate. The two utilized components in SFTA-Net enriched the tampered regions with distinguishing information. The visualization was compared with the help of masks of CASIA 2.0 dataset which contain the tampered region annotations. The SFTA-Net includes the auxiliary information lacking in SG-module-based CNN model training, resulting in improved classification performance. Similarly, the contrastive information via triplet loss steadily increased the model performance. The SG was lacking in previous studies, which instead focused on global or deep features that lack important information. Therefore, a DL-based tampered areas-based focusing approach was needed to provide a robust solution. The first column shows the input images, while the second shows the tampered area maps. These maps were not used in the SFTA-Net-based model input but were used to compare self-guided head-based tampered area detection. Lastly, column 3 shows the features of the SG head, which show tampered area-based pixel super-resolution. The model shows that the SGA head adds important information when used to identify forged or original images. Feature maps exhibit a gradual transition from 0 to 1, followed by a sharp discontinuity where values drop to 0 in saturated areas. This behavior can be attributed to activation clipping, normalization effects, or model design constraints that suppress noninformative or overly activated regions. This mechanism enhances contrast and improves feature localization, ensuring that only the most relevant information is emphasized.

Figure 3 The SG-head-based features extraction of three samples has been shown as three columns: (A) tampered images, (B) forged regions masks, and (C) SGA-Head module-based detected feature maps.

Conclusion

Data are continuously increasing over-time on the internet and other devices. The increased usage of the internet increased the amount of fake or forged data as well. Forged data include text, images, voices, and other data formats. However, detecting forged data is essential as it can be used to reduce misinformation. To identify forged digital imaging data, many previous studies have proposed using DL-models’ fusion, hand-crafted feature extraction, and transfer learning methods. However, these methods are not robust or generalized and do not add any self-supervision or guidance. To solve this problem and to provide the SG solution, this study proposed the SG head-based auxiliary and triplet loss-based CNN (SFTA-Net) model which not only identifies tampered and original images but also provides self-guidance into the images. Furthermore, the contrastive information is more generously added via triplet loss which highly improves the model performance by closing the similar samples. It is a lightweight approach with only a few convolutional layers, and leverages the auxiliary information and spatially guided head to improve forgery detection. The CASIA 1.0 and 2.0 image datasets were used for experimentation and were compared by including each component in the SFTA-Net model, which outperformed other methods and proved to be the more robust and generalized approach for copy-move and splicing forgery detection. The Cross-dataset testing proved the robustness and generalization of the proposed method more rigorously.

In the future, exploring strategies like domain adaptation or multi-domain training will be essential in order to enhance robustness and generalization across diverse real-world scenarios, instead of only focusing on two types of forgery detection.

Supplemental Information

Supplemental Information 1 Code.

Additional Information and Declarations

Competing Interests

The author declares that she has no competing interests.

Author Contributions

Amerah Alabrah conceived and designed the experiments, performed the experiments, analyzed the data, performed the computation work, prepared figures and/or tables, authored or reviewed drafts of the article, and approved the final draft.

Data Availability

The following information was supplied regarding data availability:

The data is available at Kaggle: https://www.kaggle.com/datasets/sophatvathana/casia-dataset.

The code are available in the Supplemental File.

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
