# Peer review of "SFTA-Net: a self-supervised approach to detect copy-move and splicing forgery to leverage triplet loss, auxiliary loss, and spatial attention"

_PeerJ Computer Science, doi:10.7717/peerj-cs.2803_

## Round 0.1 · original submission · Major Revisions

Dear authors,

You are advised to critically respond to all comments point by point when preparing an updated version of the manuscript and while preparing for the rebuttal letter. Please address all comments/suggestions provided by reviewers, considering that these should be added to the new version of the manuscript. In particular, R3 is very critical, and so you should carefully consider all their points

Kind regards,
PCoelho

Reviewer 1 ·

Basic reporting

The authors gives an image forgery threatens article. They presented it to spread misinformation and impact legal decisions, making effective detection vital. Traditional methods have limitations, focusing either on single forgery types or lacking tampered region detection. The proposed SFTA-Net addresses these issues with a lightweight deep-learning approach, incorporating self-guidance and auxiliary losses for better tampered region detection. Tests on CASIA 1.0 and CASIA 2.0 datasets showed SFTA-Net outperforming previous methods, achieving 95% and 97% accuracy respectively.

Also their approach enhances forgery detection and helps maintain digital trust.


In general, it is a well-prepared study. Only the literature survey is very insufficient. More studies should be presented and the state of the art should be strengthened.

Experimental design

well disegned.

Validity of the findings

well prepared.

Additional comments

In general, it is a well-prepared study. Only the literature survey is very insufficient. More studies should be presented and the state of the art should be strengthened.

Reviewer 2 ·

Basic reporting

no comment

Experimental design

no comment

Validity of the findings

no comment

Additional comments

1. Triplet Loss is contrastive loss?Both in the title and abstract, but not in the contribution. Why?
2. What does "Tempered" in Figure 1 mean? The word appears several times in the article
3. In line 200, what is a few convolution and ReLU layers?
4. In Line 212 and 213, what is Table ??
5. This method introduces commutative loss, and then auxiliary loss, but it's not clear how it's formed. Is it just two cross-entropy losses?
6. The variables in formula 8 are not defined. In the process of forensics, there is only one image to be detected. How to select anchor, positive and negative?
7. Quantitative comparisons of different methods are missing, with no visual results
8. Absence of ablation experiment

·

Basic reporting

1. The related work is not thoroughly discussed. The authors do not compare the proposed method, nor do they mention, many major works in forensics (see for instance [1—4]. While claimed novelty of the paper consists in the use of triplet loss and attention layers, the paper does not discuss existing use of both in forensics (see for instance [5-8]. Only the casia datasets are used (and not cited!) in the experiments, which are old and relatively easy to detect, even visually.
2. Citations are difficult to read in the absence of hyperlinks and enclosing parentheses
3. The paper contains many typos (for instance "tempered") and most importantly unnatural sentences, making its reading and understanding very difficult. The authors should ask a native English speaker for help with proofreading.
4. Missing internal references to a table (lines 212, 213)
5. The paper follows a standard structure, making it easy enough to follow

[1] Cozzolino, Davide, and Luisa Verdoliva. "Noiseprint: A CNN-based camera model fingerprint." IEEE Transactions on Information Forensics and Security 15 (2019): 144-159.
[2] Guillaro, Fabrizio, Davide Cozzolino, Avneesh Sud, Nicholas Dufour, and Luisa Verdoliva. "Trufor: Leveraging all-round clues for trustworthy image forgery detection and localization." In Proceedings of the IEEE/CVF conference on computer vision and pattern recognition, pp. 20606-20615. 2023.
[3] Cozzolino, Davide, Giovanni Poggi, and Luisa Verdoliva. "Efficient dense-field copy–move forgery detection." IEEE Transactions on Information Forensics and Security 10, no. 11 (2015): 2284-2297.
[4] Cozzolino, Davide, Giovanni Poggi, and Luisa Verdoliva. "Splicebuster: A new blind image splicing detector." In 2015 IEEE International Workshop on Information Forensics and Security (WIFS), pp. 1-6. IEEE, 2015.
[5] Beuve, Nicolas, Wassim Hamidouche, and Olivier Deforges. "DmyT: Dummy triplet loss for deepfake detection." In Proceedings of the 1st Workshop on Synthetic Multimedia-Audiovisual Deepfake Generation and Detection, pp. 17-24. 2021.
[6] Wan, Qian, and Qin Zou. "Learning metric features for writer-independent signature verification using dual triplet loss." In 2020 25th international conference on pattern recognition (ICPR), pp. 3853-3859. IEEE, 2021.
[7] Sun, Ke, Hong Liu, Taiping Yao, Xiaoshuai Sun, Shen Chen, Shouhong Ding, and Rongrong Ji. "An information theoretic approach for attention-driven face forgery detection." In European Conference on Computer Vision, pp. 111-127. Cham: Springer Nature Switzerland, 2022.
[8] Luo, Yuchen, Yong Zhang, Junchi Yan, and Wei Liu. "Generalizing face forgery detection with high-frequency features." In Proceedings of the IEEE/CVF conference on computer vision and pattern recognition, pp. 16317-16326. 2021.

Experimental design

1. The proposed method is inherently flawed, as the network is trained and tested on the same dataset. This is not good practice in image forensics, as it means the trained model is tied to detect images that were tampered in a very similar manner as in the dataset (a rare fact given the variety of possible manipulation, even more so when the dataset is old). It is also prone to false positives when natural images are simply different than those in the dataset.
2. The experiments are only done on the CASIA datasets. These datasets are old and not sufficient for thorough experimentation, as they are among the easiest to detect [1]. Other relevant datasets would include [2—5] as well as the OpenMFC datasets (https://mfc.nist.gov/)
3. In Table 4, several SOTA methods are not tested on both CASIA datasets, and/or are tested on different datasets. Standard practice would consist in testing all methods on the same datasets.
4. The code is provided, but not the weights. Details to reproduce the method, such as the training setting and weights initialization, are not available in the paper, only in the code.


[1] Guillaro, Fabrizio, Davide Cozzolino, Avneesh Sud, Nicholas Dufour, and Luisa Verdoliva. "Trufor: Leveraging all-round clues for trustworthy image forgery detection and localization." In Proceedings of the IEEE/CVF conference on computer vision and pattern recognition, pp. 20606-20615. 2023.
[2] Bammey, Quentin, Tina Nikoukhah, Marina Gardella, Rafael Grompone von Gioi, Miguel Colom, and Jean-Michel Morel. "Non-semantic evaluation of image forensics tools: Methodology and database." In Proceedings of the IEEE/CVF Winter Conference on Applications of Computer Vision, pp. 3751-3760. 2022.
[3] Mahfoudi, Gaël, Badr Tajini, Florent Retraint, Frederic Morain-Nicolier, Jean Luc Dugelay, and P. I. C. Marc. "Defacto: Image and face manipulation dataset." In 2019 27Th european signal processing conference (EUSIPCO), pp. 1-5. IEEE, 2019.
[4] Korus, Paweł, and Jiwu Huang. "Multi-scale analysis strategies in PRNU-based tampering localization." IEEE Transactions on Information Forensics and Security 12, no. 4 (2016): 809-824.
[5] Korus, Paweł, and Jiwu Huang. "Evaluation of random field models in multi-modal unsupervised tampering localization." In 2016 IEEE international workshop on information forensics and security (WIFS), pp. 1-6. IEEE, 2016.

Validity of the findings

Due to the issues mentioned in the experiments and lack of comparisons to major methods, the findings validity cannot be assessed. As the method trains and tests on parts of the same dataset, it is also impossible to assess whether the proposed method actually generalizes to real forgeries, and avoid false positives.

---

## Round 0.2 · Major Revisions

Dear authors,

After the previous revision round, some adjustments still need to be made. As a result, I once more suggest that you thoroughly follow the instructions provided by the reviewers to answer their inquiries clearly.

You are advised to critically respond to all comments point by point when preparing a new version of the manuscript and while preparing for the rebuttal letter. All the updates should be included in the new version of the manuscript.

Kind regards,
PCoelho

Reviewer 1 ·

Basic reporting

.

Experimental design

.

Validity of the findings

.

Additional comments

Although the literature review is still not clear enough, I find it appropriate to publish it as it is.

Reviewer 2 ·

Basic reporting

The authors have addressed my questions, so I suggest to accept it for publication.

Experimental design

The authors have addressed my questions, so I suggest to accept it for publication.

Validity of the findings

The authors have addressed my questions, so I suggest to accept it for publication.

·

Basic reporting

The literature, while still not comprehensive, is a bit better. Yet, this reviewer doubts the author's understanding of the discussed methods.
This reviewer recommends the author to read the literature very carefully, as the written information does not always correspond to what the methods actually do:
l90 "Both works depend on spatial annotations with splicing segmentation. To eliminate this dependency, a method is needed that is more adaptable to diverse datasets using self-supervision." talking about Cozzolino et al. 2015a and 2015b. This is wrong, neither methods are learning-based, thus they do not depend on spatial annotations.
l 82 "Methods like these use of Siamese networks and fusion approaches, do not incorporate auxiliary losses." The author needs to explain how a method works, not how it does **not** work, unless this is relevant to explain limitations. This reviewer understands that the author mentions existing methods do not use auxiliary losses to justify that the proposed method does it. Yet, the fact that a specific tool has not been used before is not a justification that it should be used. If all existing methods to fasten a screw involved using a screwdriver, would this be a reason to write a paper explaining that we can use a hammer to fasten a screw? To give a motivation for using a new tool, one should explain how it can improve existing methods or overcome existing limitations, rather than the simple fact it has not been used before.
Overall, this new paragraph (approximately ll 76-91) should be in the related works section rather than the introduction.
ll 16-18 "Furthermore, applied Deep Learning (DL) methods only focus on deep feature extraction without considering the focus on tampered regions detection and have not even considered any contrastive loss" See the first review for examples doing this.
This reviewer would also encourage the author to find their own references for forgery detection, including more traditional, non deep-learning based methods, that can still be very effective especially on the tested datasets.

Table 5 is very unclear.
What do precision, recall, and F1 mean when they are only compute on tampered, or on original, images? To compute these scores you need both positive and negative samples, you cannot test only on one class.
On the last row, do the authors mean "Fine-tuned on MICC-FC2000"?
Considering the results on Table 5, I assume the results on Figure 2 are when the method is trained on a part of the testing dataset, is this right? The author should make clearer, in each experiment/table entry, which datasets were seen during training/fine-tuning, and which dataset is used for testing.


The author mentioned the writing would be improved upon acceptance and before publishing using the PeerJ editing service, if this is acceptable to the editor then this reviewer will stop mentioning writing issues.

Experimental design

While the literature was slightly improved with relevant methods, these new additions are not added to the experimental comparisons, despite featuring available codes to use them.

Datasets choice
Concerning the authors' response to previous comments "While I recognize that CASIA datasets are considered relatively old, they are still widely used as benchmarks in recent forensic studies." They are indeed still used, but not alone in serious works, only in complement to more recent and thorough datasets. A method that is only trained and tested on such dataset has no practical usability. Furthermore, even with the newly-added MICC FC2000 dataset, experiments
still focus on copy-move and splicing, whereas the method is claimed to be generically against forgeries. Please see my previous review for datasets that could be used to validate the method on various aspects and manipulation types.
In this reviewer's opinion, more comprehensive experiments on different datasets should not be considered future works (as stated by the author in the response), but is indeed a crucial part for the paper to be publishable. In the first review, several datasets were suggested. While the OpenMFC ones can be considered unwieldy due to its size, the other suggested ones are much smaller and thus easy to use in experiments.

As training (or fine-tuning) and testing are done on the same datasets, can the author confirm the dataset is split in training/validation/testing, and indicate the proportions of the datasets used in each segment? The training/testing (and validation if there are hyperparameters) split is necessary for the results to have any validity.

The author added a test with testing done on another dataset than the one used for training, which goes in the right direction.

Validity of the findings

Thanks for adding the tests with testing done on another dataset than the one used for training. However, from what this reviewer understands of Table 5 both the F1 and accuracy are below 50% without fine-tuning. This means that the method does not generalize at all and is unable to detect forgeries when the image and manipulation are extremely similar to the training dataset (both in terms of initial capture and processing and of how the manipulation is done). This strongly limits its usability in practice. While results strength and impact are not assessed in this journal, the author should not oversell the method by stating that it generalizes well in such a case.

Overall, it is difficult to assess the validity of the findings as long as major points in the first two review sections are not addressed.

Additional comments

Overall, the author presented a few improvements on the paper, with more discussion on the literature, and experiments on an unseen dataset during training.
However, the paper remains inherently flawed on three aspects (see details below), and major improvements are still needed.
* The literature, while still not comprehensive, is a bit better. Yet, this reviewer doubts the author's understanding of the discussed methods. More importantly, the newly added methods are indeed discussed, but not used in the experiments to compare the proposed method.
* A new experiment on an unseen dataset was added. However, this reviewer does not understand how certain parts of the results are presented (e. g., what is a F1 score when tested only on authentic samples?). The results seem to be no better than random when the dataset has not been seen during training. This strongly limits the usability of the method. Even though potential impact is not to be assessed in this journal, this should be stated as a clear limitation of the method rather than overselling on the simple fact that "it still works if fine-tuning on the new dataset": how does one fine-tune on wild images?
* While one dataset was added to the experiments, the choice of datasets (casio and micc-f2000) is still too narrow to know whether the proposed method can work on forgeries other than splicing and copy-move, especially on forgeries seen nowadays.

Other comment:
l100 "A contrastive loss namely Triplet loss": Triplet loss is not a contrastive loss, these are similar but distinct concepts

---

## Round 0.3 · Minor Revisions

Dear author,

Thanks a lot for your efforts to improve the manuscript.

Once more, please address the minor comments presented by the reviewer, particularly the following, that are critical:

- The results from other methods were reported as stated in the original papers rather than based on experiments from the author (meaning that SOTA methods, which do not report results on the databases used by the author, are not shown in the experiments). The author states this limitation is due to cost reasons), which weakens the results. Nevertheless, it may be acceptable if this is stated and justified in the manuscript.

- Additionally, it is advisable to use some professional language services to improve the quality of the manuscript.

Kind regards,
PCoelho

·

Basic reporting

The global reporting is much better, though I strongly recommend the use of a professional copy-editing service to improve the language.

Tables and Figures captions are still a bit short. They should include not only the direct information as to what is presented, but also explain briefly what conclusions can be drawn from the figure. Ideally, a reader should be able to understand the main ideas and results of the paper by looking at the figures and tables first.

Figure 3 the use of a colored background in the table is unsettling and probably unnecessary.

Line 168 the method is called "Noiseprint", not "noise-print"

Experimental design

I understand the concern about a computational cost to run the existing SOTA and reproduce the results. While non-standard and unfortunately limiting to assess the validity of the results (especially as it means methods that did not report results on a given dataset do not appear in the table), I am willing to accept this, as long as this is fine with the editor. However, this limitation should be clearly stated in the paper.

Validity of the findings

Figure 3 I would like to see discussed why the feature map seems to go from 0 to 1 gradually, then directly to 0 with a sharp discontinuity, in the saturated areas.

Table 6 I am mostly satisfied with the rebuttal. On the table, I would like to see a column indicated for each entry on which dataset the method is trained/fine-tuned. We should have two entries for the MICC-FC2000 dataset, one showing the results without fine-tuning, and one with fine-tuning. This would show why fine-tuning is necessary with this method. Being clear on this is very important as it is a strong limitation of the method, as it prevents its use in the wild, or in all scenarios where example images with ground truths are not available.

Additional comments

Most of my concerns were addressed, and I am ready to accept the paper with minor revisions, mainly 1. make it clear that the results from other methods were as reported in the original paper rather than reproduced, 2. clarify Table 6 as suggested, and 3. make use of a copy-editing service to improve the language before the final publication

---

## Round 0.4 · accepted · Accept

Dear authors, we are pleased to verify that you meet the reviewer's valuable feedback to improve your research.

Thank you for considering PeerJ Computer Science and submitting your work.

Kind regards
PCoelho